# Cortical Plasticity and Interneuron Recruitment in Adolescents Born to Women with Gestational Diabetes Mellitus

**DOI:** 10.3390/brainsci11030388

**Published:** 2021-03-19

**Authors:** Jago M. Van Dam, Mitchell R. Goldsworthy, William M. Hague, Suzette Coat, Julia B. Pitcher

**Affiliations:** 1Robinson Research Institute, Adelaide Medical School, University of Adelaide, Adelaide, South Australia 5005, Australia; jago.vandam@adelaide.edu.au (J.M.V.D.); bill.hague@adelaide.edu.au (W.M.H.); suzette.coat@adelaide.edu.au (S.C.); 2Hopwood Centre for Neurobiology, Lifelong Health Theme, South Australian Health and Medical Research Institute (SAHMRI), Adelaide, South Australia 5000, Australia; 3Obstetric Medicine, Women’s and Children’s Hospital Network, North Adelaide, South Australia 5006, Australia; 4Institute for Mental and Physical Health and Clinical Translation, School of Medicine, Deakin University, Geelong, Victoria 3220, Australia

**Keywords:** gestational diabetes, transcranial magnetic stimulation, cortisol, neuroplasticity, I-waves, neurodevelopment

## Abstract

Exposure to gestational diabetes mellitus (GDM) in utero is associated with a range of adverse cognitive and neurological outcomes. Previously, we reported altered neuroplastic responses to continuous theta burst stimulation (cTBS) in GDM-exposed adolescents. Recent research suggests that the relative excitability of complex oligosynaptic circuits (late I-wave circuits) can predict these responses. We aimed to determine if altered I-wave recruitment was associated with neuroplastic responses in adolescents born to women with GDM. A total of 20 GDM-exposed adolescents and 10 controls (aged 13.1 ± 1.0 years) participated. cTBS was used to induce neuroplasticity. I-wave recruitment was assessed by comparing motor-evoked potential latencies using different TMS coil directions. Recruitment of late I-waves was associated with stronger LTD-like neuroplastic responses to cTBS (*p* = < 0.001, *R*^2^ = 0.36). There were no differences between groups in mean neuroplasticity (*p* = 0.37), I-wave recruitment (*p* = 0.87), or the association between these variables (*p* = 0.41). The relationship between I-wave recruitment and the response to cTBS previously observed in adults is also present in adolescents and does not appear to be altered significantly by in utero GDM exposure. Exposure to GDM does not appear to significantly impair LTD-like synaptic plasticity or interneuron recruitment.

## 1. Introduction

Gestational diabetes mellitus (GDM) affects approximately 10% of pregnancies, with higher prevalence in overweight and obese women [1]. Evidence indicates that children exposed in utero to GDM are at higher risk of neurodevelopmental difficulties, including attention deficit hyperactivity disorder [2], autism spectrum disorders [3], and impaired motor development [4]. Recently, we reported [5] that children born to women with GDM had highly variable long-term depression (LTD)-like neuroplastic responses to magnetic brain stimulation. In our study, children’s cortical plasticity was associated with the severity of the mother’s GDM, particularly with the degree of maternal insulin resistance both before and during GDM treatment. These results were consistent with animal research suggesting that oxidative stress and inflammation associated with maternal hyperglycaemia (resulting from insulin resistance) are major drivers of altered neurodevelopment in GDM-affected fetuses [6]. Both human and animal research suggests that GDM-exposed fetuses experience an adverse environment in utero that contributes to abnormal neurodevelopment, possibly including altered synaptic plasticity and excitability. However, to our knowledge, ours was the first study to examine neurophysiological processes in humans exposed to GDM.

We previously used a form of repetitive transcranial magnetic stimulation (TMS), continuous theta burst stimulation (cTBS), to induce and measure LTD-like neuroplasticity. cTBS, along with its counterpart intermittent TBS (iTBS), induces after-effects on excitability in the stimulated area that outlast the period of stimulation by minutes to hours [7]. Evidence suggests that these effects depend on the activity of N-Methyl-D-Aspartate (NMDA) receptors, and hence it is thought that they represent an analogue of early stages of synaptic plasticity in the human brain [8,9]. However, a principal issue with neuroplasticity induction by brain stimulation is that the responses are often highly variable both within and between individuals, with a number of known contributing factors, including age, cortisol, genetics, and prior muscle activity [10]. Recent research [11,12,13] suggests that inter-individual differences in the cortical network activated by TMS pulses can also influence the response. Because different populations of cortical neurons are stimulated more easily or are more excitable in different people at different times and because these populations can contribute differently to the neuroplastic response, a substantial proportion of the variability in neuroplastic responses may be explained by variable interneuron recruitment, rather than by variability in synaptic function *per se*.

The descending volley evoked by a single TMS pulse consists of several components. The earliest of these is believed to reflect the direct activation of the corticospinal output cells and is known as the direct (D)-wave. The later components result from indirect activation of output cells and are known as indirect (I)-waves. Early I-waves likely result from monosynaptic input to corticospinal neurons from layer II/III interneurons, whereas later I-waves reflect the activity of more complex, oligosynaptic circuits [13,14].

Hamada, Murase, Hasan, Balaratnam, and Rothwell [11] found that, in their sample, there was no net facilitatory or inhibitory neuroplastic response to iTBS or cTBS, respectively, due to high response variability. As is common in such experiments, there was variability not only in the magnitude of neuroplastic responses but also in the direction of responses. That is, some individuals exhibited inhibitory responses to iTBS, or facilitatory responses to cTBS—the opposite to what is expected. However, individuals in whom TMS recruited later I-waves had stronger neuroplasticity and, when treated as a group, displayed significant neuroplastic responses to both iTBS and cTBS in the expected directions. Similar results were observed by Hordacre, Goldsworthy, Vallence, Darvishi, Moezzi, Hamada, Rothwell, and Ridding [13]. Although the mechanisms are not entirely clear, Hamada, Murase, Hasan, Balaratnam, and Rothwell [11] suggest that the late I-wave-generating circuitry may be more sensitive to TBS than the early I-wave-generating circuitry.

Given that we previously observed both weaker and more variable neuroplastic responses to cTBS (including more LTP-like responses) in children exposed to GDM when compared with a control group, we aimed in the present study to determine whether I-wave recruitment differed in this group and whether this could explain the increased variability in neuroplastic responses we observed. Given that the association between I-waves and neuroplasticity has only been studied in adults, our results also provide novel evidence for the presence of this effect in adolescents.

## 2. Materials and Methods

### 2.1. Ethical Approval

All procedures were approved by the Women’s and Children’s Health Network and University of Adelaide human research ethics committees (protocol code HREC/16/WCHN/50; date of approval 08/06/2016) and conducted in accordance with the Declaration of Helsinki (2008 revision). Participants were pre-screened for contraindications to TMS [15]. Parents and participants provided written, informed consent. Parents accompanied their children to the experimental session.

### 2.2. Subjects

A total of 20 GDM-exposed subjects (aged 12.7 ± 0.8 years (mean ± SD), 10 female), all of whom participated in our previous study [5], were recruited from the Adelaide arm of the Metformin in Gestational diabetes (MiG) trial [16]. Their mothers had been treated for GDM not responding to lifestyle alteration, with a 1:1 random allocation at study entry to receive either insulin (*n* = 12) or metformin treatment (*n* = 8). Gestational age at trial entry varied from 20 weeks to 33 weeks (mean 30 ± 3.4 weeks). Three of the metformin group received supplementary insulin to achieve maternal euglycaemia. Ten control participants (aged 13.6 ± 1.3 years, 7 females) whose mothers had normal glucose tolerance on routine testing at the end of the second trimester and no other recorded major pregnancy complications were recruited from labour ward records and matched as closely as possible for age, sex, gestational age at birth, and mode of delivery.

### 2.3. Electromyography

Participants were seated with their hands and forearms supported. Adhesive Ag/AgCl bipolar surface electrodes were applied over the right first dorsal interosseous (FDI) hand muscle to obtain surface electromyography (EMG) recordings. EMG signals were amplified (×1000; 1902 amplifier; CED), bandpass filtered (20 Hz–1 kHz), and digitized at 5 kHz (1401 interface; CED) and were stored offline for later analysis.

### 2.4. Transcranial Magnetic Stimulation (TMS)

TMS is a non-invasive brain stimulation technique in which the motor cortex is electromagnetically stimulated to produce a motor evoked potential (MEP), recorded in a contralateral muscle using EMG [17]. Motor cortical excitability was assessed with single-pulse TMS applied to the left primary motor cortex (M1) representation of the right FDI muscle using a 70 mm figure-of-eight coil connected to a monophasic Magstim 200^2^ stimulator (Magstim Co, Whitland, UK). Previous research has shown clearly that different descending volleys are elicited by single-pulse TMS depending on the direction of current flow across the motor cortex. Posterior-anterior (PA) current preferentially elicits early I-waves, whereas anterior-posterior (AP) current can recruit late I-waves. High-intensity latero-medial (LM) current evokes D-waves [11,14].

The protocol used here is equivalent to that described by Hamada, Murase, Hasan, Balaratnam, and Rothwell [11]. Three different coil orientations were used to evoke MEPs: (1) PA-directed currents were produced by the coil held posterolaterally at an angle of about 45° to the midline, (2) AP-directed currents were elicited by placing the coil 180° to the PA current position, and (3) LM-directed currents were produced with the coil placed leftwards (90° from midsagittal line).

Using PA currents, the optimal site for consistently evoking MEPs in the FDI was determined and marked on the scalp, and resting motor threshold (RMT) was determined as the lowest TMS intensity required to evoke MEPs of at least 50 µV peak-to-peak amplitude in the resting FDI in at least five of ten consecutive trials. The active motor threshold (AMT) was defined as the lowest intensity to evoke an MEP of 200 μV, or visibly distinct from background EMG activity, in more than 5 of 10 consecutive trials while subjects maintained approximately 10% of maximum voluntary contraction of the target muscle, measured and monitored using a digital oscilloscope. We measured AMT with PA, AP, and LM currents (AMTpa, AMTap, and AMTlm, respectively). The TMS intensity that evoked MEPs of ~1 mV peak-to-peak amplitude (SI_1mV_) was also determined, using PA current, and used throughout each experiment for evoking test MEPs [18]. SI1mV was recorded 15 min after AMT measurements, following a period of muscle relaxation. Intensities are expressed as % of maximum stimulator output (MSO). All measurements were performed at the hotspot determined by PA currents, as previous research has shown that the direction of current does not influence the position of the hotspot [19,20].

### 2.5. Measurement of MEP Onset Latency to Assess I-Wave Recruitment

The level of late I-wave recruitment was quantified by measuring MEP onset latencies using AP current in pre-contracted muscle at near-threshold stimulus intensities. AP MEP latencies were compared with MEP latencies using PA and LM currents. The rationale is that, in individuals in whom late I-wave generating circuitry is more excitable, AP MEPs will have longer latencies than PA and LM MEPs, because more complex circuits are involved in stimulating output cells (with AP current) [11,14]. Indeed, AP MEP latencies are typically observed to be 1–2 ms longer than PA MEPs, and 4–5 ms longer than LM MEPs [21]. As TBS preferentially activates the brain during the second depolarising phase of the current, that is, with an anterior-posterior current [22], those individuals who exhibit longer AP MEP latencies will be more strongly affected by TBS and show stronger neuroplastic responses [11]. The difference between LM and AP latencies (AP-LM) was used as the primary index of late I-wave recruitment because LM latencies are highly consistent, allowing reliable estimation of I-wave recruitment (larger AP-LM values indicating later I-waves).

We measured onset latencies of MEPs using PA, AP, and LM currents with stimulus intensities of 110% AMTpa, 110% AMTap, and 150% AMTlm (or 50% MSO in subjects whose 150% AMTlm did not reach 50% MSO) [11]. Relatively high stimulus intensities were used with LM current to ensure a D-wave was evoked [23]. Twenty MEPs each with PA and AP currents, and 10 MEPs with LM currents, were recorded. After every tenth trial, subjects were asked to relax their hand to minimise fatigue. Signal (Version 6; Cambridge Electronic Design) was used to record, process, and analyse EMG data. A custom-made script was used to measure MEP-onset latency, defined as the time-point where rectified EMG signals exceed a mean ± 2 SD of the pre-stimulus EMG level, expressed as milliseconds after the stimulus. For each subject and coil orientation, mean latencies were calculated and inspected directly to ensure validity.

### 2.6. LTD-Like Neuroplasticity Induction with cTBS

cTBS was used to induce LTD-like suppression of MEP amplitudes. Pharmacological studies indicate that cTBS-induced MEP suppression is NMDA-receptor-dependent and similar mechanistically to LTD [8]. An air-cooled figure-of-eight coil (rTMS coil), connected to a Magstim Super Rapid stimulator (Magstim, Whitland, UK), was used to apply repetitive TMS to the optimal site for stimulating the right FDI. The cTBS protocol consisted of 600 pulses applied in bursts of three pulses at 50 Hz, repeated at 5 Hz for a total of 40 s [7]. Stimulation intensity was set to 70% of RMT (measured using the rTMS coil). MEPs were recorded in blocks of 15 trials prior to cTBS (i.e., baseline) and at 0, 5, 10, 20, and 30 min following cTBS with PA current. In each block, the mean of the peak-to-peak amplitudes of the 15 MEPs was calculated. Changes in MEP amplitude relative to baseline MEP amplitude were used as an index of neuroplasticity [24]. All MEPs were recorded at high gain and any with obvious EMG activity in the 200 ms before the TMS stimulus were discarded. cTBS was performed following a 15-min rest period after baseline MEP recording to avoid metaplastic effects of prior cortical and muscle activity on neuroplastic response [25,26,27].

The coefficient of variation (COV) of baseline MEP amplitudes was also calculated to determine correlation with plasticity induction [13].

### 2.7. Salivary Cortisol

Saliva samples were obtained from each child immediately before TMS baseline measures (at 13:22 ± 0.25 h) using a Salivette (Sarstedt) designed for cortisol analysis. Salivettes were centrifuged to obtain saliva, which was stored at −20 °C until assayed. Twenty-five microlitre aliquots of saliva were assayed in duplicate for cortisol concentrations by enzyme-linked immunosorbent assay (ELISA) according to manufacturer instructions (HS-Cortisol; Salimetrics).

### 2.8. Statistical Analysis

Data were analysed using Python 3.7. Data were checked for normality using the Shapiro–Wilk test and were tested for equal sphericity and variance where appropriate using Bartlett’s and Levene’s tests, respectively. The associations between continuous variables (e.g., AP-LM latency and mean neuroplastic response) were tested using linear regression models. Interaction terms were added to regression models to test for group differences in the effect of AP-LM latencies (Group*AP-LM interaction) when predicting neuroplasticity. Group means were compared using an independent samples *t*-test (or a Welch’s *t*-test where appropriate). To further examine the influence of I-wave recruitment on post-cTBS MEP amplitudes, data were analysed based on the presence or absence of late I-waves by median split on AP-LM latency, where AP-LM latency differences greater than the group median were considered to represent late I-waves, and those lower to represent early I-waves [13]. The analysis was also performed based on a split at 4 ms AP-LM difference [11]. Results were comparable, so only the median split is reported below. Data were also analysed by the direction of neuroplastic response (inhibitory, <100% baseline; or facilitatory, >100% baseline) to examine the difference in I-wave recruitment.

## 3. Results

### 3.1. Baseline Characteristics and Cortical Excitability

The mean (± SD) resting motor threshold (RMT) of all participants was 47.4 ± 7.3 (% MSO). RMT did not differ significantly between the GDM (46.6 ± 6.0) and control (49.1 ± 9.6) groups (*p* = 0.379). Within the GDM group, individual RMTs in the present study correlated strongly with RMTs assessed previously [reported in 5] (*p* < 0.001, *R*^2^ = 0.574). RMT was not influenced by age (*p* = 0.251), sex (*p* = 0.563), gestational age at birth (*p* = 0.967), or maternal GDM treatment type (*p* = 0.738). Gestational age at birth did not differ between the GDM (38.8 ± 1.5 weeks) and control (39.3 ± 0.86 weeks) groups (*p* = 0.358; grand mean 39.0 ± 1.3 weeks).

Neuroplastic responses measured in the previous study were not correlated to those measured in the present study (*R*^2^ = 0.069, *p* = 0.279). In contrast to the previous study, maternal insulin resistance was not associated with the neuroplastic response (*R*^2^ = 0.116, *p* = 0.233). However, lower insulin (*R*^2^ = 0.286, *p* = 0.033) and maternal C-peptide concentrations (*R*^2^ = 0.681, *p* < 0.001) at trial entry were strongly correlated with stronger LTD-like neuroplastic responses.

### 3.2. I-Wave Recruitment Predicts the Neuroplastic Response to cTBS

Among all participants there was not a net neuroplastic response to cTBS; in a repeated measures ANOVA on raw MEP amplitudes there was no main effect of Time (*p* = 0.126). This was due to the large variability in responses, with both LTD- and LTP-like effects observed in different participants (Figure 1a). The mean post-cTBS MEP amplitude (% baseline) at the group level was 107 ± 29.

Mean (± SD) AP-LM latency difference among all participants was 3.55 ± 1.04 ms. AP-LM was strongly associated with mean neuroplastic response to cTBS, explaining 36% of the variation in post-cTBS MEP amplitudes (β = −16.817, *R*^2^ = 0.36, *p* = < 0.001) (Figure 1b). Individuals in whom late I-waves were recruited (longer AP-LM difference) were more likely to show inhibitory responses to cTBS, while individuals with shorter AP-LM differences tended to exhibit facilitatory responses. Similarly, participants in the early I-waves group, as determined by a median split on AP-LM difference, had significantly larger MEP amplitudes (123 ± 21% baseline) following cTBS than did individuals in the late I-waves group (91 ± 27) (*p* = 0.002) (Figure 2). The late I-waves group did not exhibit a significant neuroplastic response to cTBS (compared with baseline) (*p* = 0.458). However, the early I-waves group exhibited a response to cTBS consistent with significant facilitation of MEPs (*p* = 0.006). Individuals who exhibited facilitation-like responses to cTBS had significantly shorter AP-LM latency differences (2.94 ± 0.67 ms) than those who exhibited inhibition-like responses (4.36 ± 0.89 ms) (*p* < 0.001).

### 3.3. Group Comparisons of I-Waves and Neuroplasticity

Mean post-cTBS MEP amplitudes (% baseline) did not differ between the GDM (104.0 ± 28.0) and control (114.4 ± 31.5) groups (*t*_(28)_ = −0.922, *p* = 0.364) (Figure 3b), including when correcting for cortisol concentration (*p* = 0.424). Similarly, there was no effect of Group (*p* = 0.667) and no Group*Time interaction (*p* = 0.945) in a repeated measures ANOVA of MEP amplitudes. Mean AP-LM latency differences did not differ between the GDM (3.53 ± 0.79 ms) and control (3.61 ± 1.48 ms) groups (Welch’s *t*_(28)_ = −0.205, *p* = 0.869) (Figure 4a). However, there did appear to be a greater spread of AP-LM differences in the control group, with more extreme minimum and maximum values (Figure 4b), despite the smaller number of participants in this group. Levene’s test indicated a significant difference in variances in AP-LM between groups (*p* = 0.042).

The association between I-wave recruitment and neuroplasticity appeared stronger in the control group, with AP-LM latency difference explaining 86% of the variation in post-cTBS MEP amplitudes (β = −19.7751, *p* < 0.001, *R*^2^ = 0.858) in this group compared with 13% in the GDM group (β = −12.5061, *p* = 0.13, *R*^2^ = 0.125).

Despite the non-significant result in the GDM group, the association between AP-LM and plasticity was visually present (Figure 3a), but appeared to be weakened statistically by a greater proportion of AP-LM values clustered between approximately 3.5–4 ms (i.e., on the border of late I-waves), which appears to be a region where the predictive value is low regarding neuroplastic response.

The difference in magnitude of the AP-LM and neuroplasticity association between groups was not statistically significant, as there was not a significant Group*AP-LM interaction when predicting post-cTBS MEP amplitudes (Group, *p* = 0.20; AP-LM, *p* < 0.001; Group*AP-LM, *p* = 0.409). Neither the response to cTBS (*p* = 0.344) nor mean AP-LM latency (*p* = 0.162) differed between the maternal GDM treatment groups (metformin or insulin).

Resting motor threshold did not differ between the GDM (45.6 ± 6.0% MSO) and control (49.1 ± 9.6% MSO) groups (*p* = 0.379).

### 3.4. Cortisol

The mean (± SD) salivary cortisol concentration at the time of testing was 2.12 ± 1.43 nmol/L. The GDM group (2.46 ± 1.59) had higher but more variable cortisol values compared with the control group (1.39 ± 0.50), but there was no difference in mean cortisol concentration (*t*_(27)_ = 1.957, *p* = 0.06, ω^2^ = 0.09).

The control group completed the TMS experiment later in the day (IQR: 14:30–16:30 h) compared with the GDM group (IQR: 10:45–12:50 h) (*t*_(26)_ = −3.987, *p* < 0.001). However, time of test did not impact cortisol values (β = −0.002, *R*^2^ = 0.09, *p* = 0.118). Neither time (*p* = 0.471) nor group (*p* = 0.687) predicted cortisol concentration in an ANCOVA model (*F*_(2,25)_ = 1.662, Adj. *R*^2^ = 0.047, *p* = 0.238). There was no sex difference in cortisol concentrations (Welch’s *t*_(23.7)_ = −1.285, *p* = 0.211), and maternal treatment (metformin or insulin) did not affect cortisol (*p* = 0.485). The cortisol values in both the GDM and control groups were consistent with normative data for this age group and for the time of day of assessment [28].

Cortisol did not predict mean post-cTBS MEP amplitudes (*F*_(27,1)_ = 2.254, *R*^2^ = 0.08, *p* = 0.128) or RMT (β = −1.595, *R*^2^ = 0.095, *p* = 0.095).

There was a near-significant association between cortisol and AP-LM latency difference, whereby higher cortisol concentrations were associated with greater AP-LM difference (β = 0.243, *R*^2^ = 0.134, *p* = 0.051), but cortisol as a covariate did not alter any of the associations between group, AP-LM, and response to cTBS described above.

## 4. Discussion

We have provided evidence indicating that the association between interneuron recruitment and the neuroplastic response to cTBS, whereby individuals who exhibit motor responses produced by late I-wave-generating circuitry have stronger and more predictable neuroplastic responses, is present in adolescents to a similar degree to that previously observed in adults [11,12,13]. Although I-wave recruitment was less variable in GDM-exposed participants, mean MEP latencies were similar. The association between I-wave recruitment and the neuroplastic response in this group appeared weaker, but this difference was not statistically significant. Thus, it does not appear that *in utero* GDM exposure fundamentally alters the relationship between I-wave recruitment and the neuroplastic response to cTBS.

In the present study we did not observe a net, group-level neuroplastic response to cTBS. Given the variability often observed in the response to cTBS [11,13,29,30], this was not entirely unexpected and largely reflects variability in the direction of responses. That is, whereas some individuals exhibit the expected suppression-like response, often a similar number exhibit responses consistent with facilitation of MEPs, such that the group average post-cTBS MEP amplitude is near baseline. However, in contrast to our previous study [5], we did not observe any systematic difference in response to cTBS in the GDM group compared with the control group. There are several possible explanations for this result. Firstly, given the responses observed in the current control group, our previous control group may have been particularly responsive to cTBS. That is, by chance we may have recruited subjects in the initial study who exhibited particularly robust LTD-like responses to cTBS, resulting in inflation of the difference between groups. Nevertheless, we do believe our original conclusion—that there is an effect of GDM on child neurophysiology—to be valid, because there was a range of supporting evidence within the GDM group, including associations between child neurophysiology and maternal insulin resistance during pregnancy [5]. However, the effect size in the group comparison is likely smaller than originally estimated, and it may require a larger sample size to make reliable conclusions.

There may also be an effect of age: as our GDM group subjects have aged approximately two years since first assessment, and given the rapid stage of their development, the neurophysiological effects we observed previously may have normalised and may now be relatively subtle, if still present.

Given these results, we were unlikely to find support for our original hypothesis that differential I-wave recruitment would explain the difference in neuroplasticity between groups that we previously observed. Indeed, the mean late I-wave recruitment (AP-LM latency) was similar in the GDM group when compared with controls. However, there was an apparent difference in the distribution of I-waves, as the GDM group, despite having a larger sample, had a smaller range of responses, with both smaller group minimum and maximum AP-LM latency differences. The effect of this was a clustering of AP-LM latencies around the 4 ms mark, which represents approximately the border between early and late I-waves [11,21]. Responses in this range appear to have lower predictive value of the neuroplastic response to cTBS, and the association between I-wave recruitment and neuroplastic response therefore appeared weaker (and non-significant) in the GDM group when compared with the robust association seen in the control group. However, this comparison did not prove to be significant, and we must therefore conclude that there is no difference in interneuron recruitment by TMS or in the relationship between late I-wave recruitment and neuroplastic responses to cTBS, in children exposed to GDM, unless such an effect is smaller than we can detect with our sample size. Although the control group was small, the effect size therein was extremely strong. Thus, a larger control group would almost certainly result in a reduction in the magnitude of the difference between groups; therefore, the sample is biased towards a false positive, so we can accept a negative result with relative confidence.

The precise mechanisms underlying the inter-individual differences in late I-wave recruitment are unclear and likely complex. Presumably, individuals differ with respect to the relative excitability of early and late I-wave-generating circuitry and, consequently, their responsivity to cTBS. Given that the recruitment of late I-waves is highly consistent within individuals [11], this excitability bias must be relatively fixed. That is, individuals who do not recruit late I-waves should have consistently lower excitability of late I-wave-generating circuitry. Alternatively, these individuals may have structural differences in neural circuitry in the motor cortex, possibly possessing a lower proportion of complex, oligosynaptic inputs to pyramidal cells. In any case, I-wave recruitment, given its reliability, is likely not the primary driver of cTBS response because this would necessitate that similarly consistent neuroplastic responses be observed; test–retest reliability of TBS is relatively modest, especially when not accounting for circadian patterns [29,31,32,33], and indeed, we did not find a correlation between neuroplastic responses measured here and in our previous study. Thus, variation in the association between I-wave recruitment and response to cTBS may explain the apparent (albeit non-significant) difference in this relationship between groups seen in this study, rather than any intrinsic difference in physiology. Nevertheless, future research may extend this analysis with a larger sample to determine whether GDM-exposed individuals exhibit differences in the excitability or structure of I-wave-generating circuitry.

It is worth noting that there are a number of alternative methods by which one can probe I-waves and their recruitment, including direct (invasive) measurement from electrodes in the corticospinal tract and one that utilises interstimulus intervals derived after constructing short interval intracortical facilitation (SICF) curves [34]. Our rationale for using the AP-LM method was twofold; it is not invasive, and secondly, it takes significantly less time than the SICF method, which is an important consideration in a cohort of children who have undergone a number of other investigations in addition to the one described here.

In contrast to our previous study [5], we did not observe lower cortisol concentrations in the GDM group. Although mean cortisol was higher in the GDM group, median cortisol was similar in each group, and the GDM group had higher variability and several large values, possibly reflecting stress responses not observed in the control group. This variability may also explain the lack of associations between cortisol and both neuroplasticity and cortical excitability that have been observed previously [5,24]. In the present study, we observed clear trends following the expected patterns, so a lack of statistical power is a likely explanation. Nevertheless, a single afternoon cortisol measurement is not sufficient to reliably characterise hypothalamic-pituitary-adrenal activity.

## 5. Conclusions

Although our results surprised us in several ways, they may be considered positive: they suggest that GDM, when properly diagnosed and treated, does not result in significant dysfunction in cortical excitability or in the forms of synaptic plasticity measured using cTBS lasting into adolescence. Further, maternal treatment with metformin, a drug that crosses the placenta [35], was not associated with adverse outcomes, supporting its safety and efficacy in treating GDM with respect to child neurodevelopment [36,37,38]. However, given our past findings regarding maternal insulin resistance and child neurophysiological outcomes [5] and the known functional neurological outcomes in individuals exposed to GDM (e.g., [2,3,4]), improvements to prevention, screening, diagnosis, and treatment practices for GDM remain important areas of investigation. Further, an analysis of forms of facilitatory synaptic plasticity may be warranted. Questions regarding the possible physiological mechanisms underlying the functional neurological consequences of GDM remain open and should be investigated further.

## Figures and Tables

**Figure 1 brainsci-11-00388-f001:**
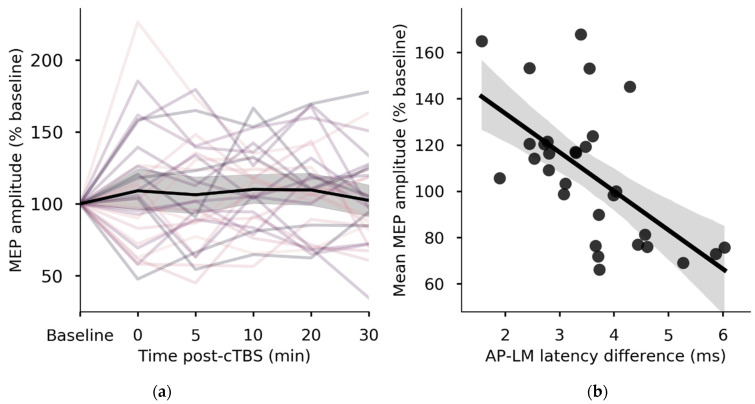
Continuous theta burst stimulation (cTBS) response and I-wave recruitment latencies. (**a**) Individual responses to cTBS varied in magnitude and direction, while the group mean (black line) remained near baseline. Shaded area represents 95% CI for the group mean. (**b**) Greater late I-wave recruitment, as indicated by longer anterior-posterior (AP)- latero-medial (LM) latency difference, was strongly associated with the magnitude and direction of the neuroplastic response to cTBS (*p* < 0.001, *R*^2^ = 0.36).

**Figure 2 brainsci-11-00388-f002:**
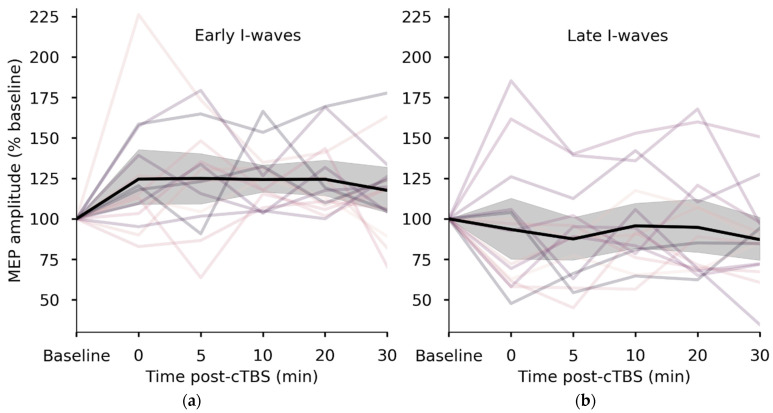
cTBS response by I-waves group. Mean post-cTBS MEP amplitudes were lower in individuals who recruited late I-waves (median split on AP-LM latency difference; *p* = 0.002). (**a**) In the early I-waves group, the response was consistent with facilitation of motor evoked potentials (MEPs) (*p* = 0.006). (**b**) In the late I-waves group, there was no group-level effect of cTBS, but most individuals showed inhibition-like responses.

**Figure 3 brainsci-11-00388-f003:**
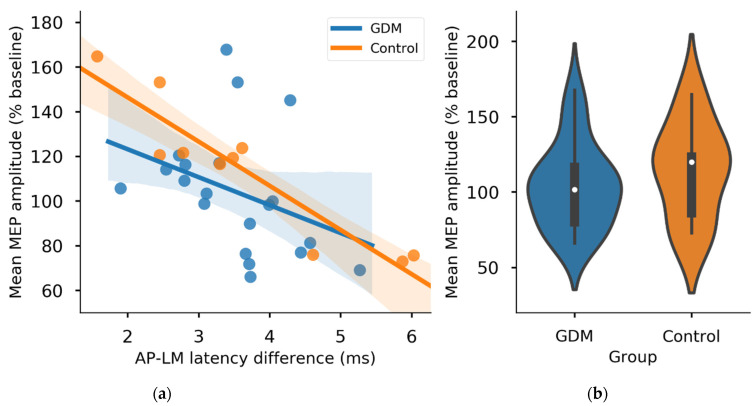
Comparison of cTBS response and I-waves-plasticity association by gestational diabetes mellitus (GDM) group. (**a**) The association between I-wave recruitment and response to cTBS appeared stronger in the control group (*p* < 0.001, R^2^ = 0.858) than in the GDM group (*p* = 0.13, *R*^2^ = 0.125), but this difference was not statistically significant (Group*AP-LM interaction, *p* = 0.41). (**b**) Mean post-cTBS MEP amplitudes did not differ between groups (inner boxplots). Shaded areas on the scatterplot represent 95% confidence intervals for the regression lines.

**Figure 4 brainsci-11-00388-f004:**
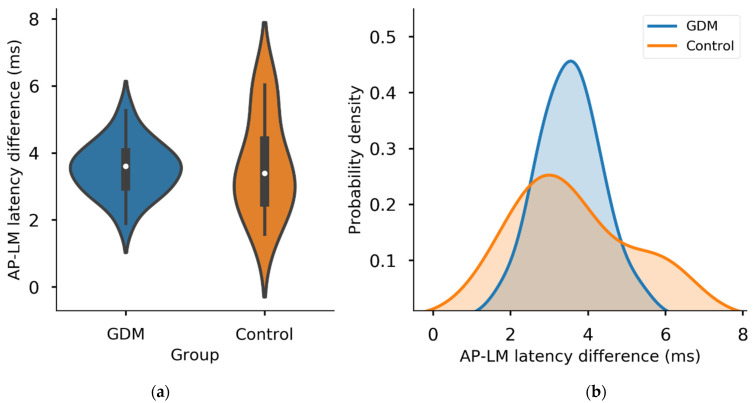
I-wave latencies by GDM group. (**a**) The central tendency of AP-LM latency differences did not differ between groups, but (**b**) latencies were more variable in the control group (*p* = 0.042; kernel density estimation plot).

## Data Availability

The data presented in this study are available on request from the corresponding author. The data are not publicly available due to privacy and ethical reasons.

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
