# Peer review of "Cortical Plasticity and Interneuron Recruitment in Adolescents Born to Women with Gestational Diabetes Mellitus"

_brainsci, 2021, doi:10.3390/brainsci11030388_

Round 1

Reviewer 1 Report

In their manuscript, Van Dam et al., studied I-wave recruitment in adolescents born to women with gestational diabetes mellitus. Their main finding was that these adolescents did not differ in the evaluated measures from healthy controls. Overall, the manuscript is well-written and logical. The results are of interest to the readers of Brain Sciences.

  • Although the authors have applied a previously used paradigm to evaluate the I-waves, I think this method is still the least established method and the authors might want to consider adding this to the discussion. The more established method to differentiate early and late I-waves is with paired-pulse TMS. When relying on the method using different coil angles, as done in the current study, the results are susceptible to more variability as the precentral gyrus possibly has a different angle between the subjects.
  • In addition to evaluation corticospinal excitability related plasticity, the study design also allows evaluating corticospinal inhibition related plasticity via silent periods, as the MEPs were induced during muscle contraction. The authors might want to consider adding results from silent periods as it would increase the value of the manuscript.
  • In how many subjects did the authors use 50% of maximum stimulator output as the stimulation intensity?
  • Part of the text may be copied from somewhere else as authors refer to “Chapter 5” (page 10, line 416) which does not exist in the current manuscript. Please check the text.
  • Please add the sections in the end from author contributions to conflicts of interest.

Author Response

Reviewer 1:

In their manuscript, Van Dam et al., studied I-wave recruitment in adolescents born to women with gestational diabetes mellitus. Their main finding was that these adolescents did not differ in the evaluated measures from healthy controls. Overall, the manuscript is well-written and logical. The results are of interest to the readers of Brain Sciences.

  • Although the authors have applied a previously used paradigm to evaluate the I-waves, I think this method is still the least established method and the authors might want to consider adding this to the discussion. The more established method to differentiate early and late I-waves is with paired-pulse TMS. When relying on the method using different coil angles, as done in the current study, the results are susceptible to more variability as the precentral gyrus possibly has a different angle between the subjects.

We presume that the reviewer is referring to the technique employed by Opie et al (Brain Sci. 2021, 11(1), 121), using short interval intracortical facilitation (SICF). The method also uses multiple coil orientations, so is unlikely to preclude the variability described by the reviewer. The alternative is to use direct recording from the corticospinal tract, which is highly unlikely to have been given ethical approval. We have included a paragraph in the discussion (Page 11, lines 435-432) acknowledging that the AP-LM method is not the only one.

  • In addition to evaluation corticospinal excitability related plasticity, the study design also allows evaluating corticospinal inhibition related plasticity via silent periods, as the MEPs were induced during muscle contraction. The authors might want to consider adding results from silent periods as it would increase the value of the manuscript.

We thank the reviewer for the suggestion, but evaluating corticospinal silent periods is not possible because the post-MEP recording epoch was too short.

  • In how many subjects did the authors use 50% of maximum stimulator output as the stimulation intensity?

We presume the reviewer is referring to the intensity used to evoke MEPs in the AP position. Two GDM participants were tested at 50% MSO.

  • Part of the text may be copied from somewhere else as authors refer to “Chapter 5” (page 10, line 416) which does not exist in the current manuscript. Please check the text.

This has been corrected.

  • Please add the sections in the end from author contributions to conflicts of interest.

These have been included.

Reviewer 2 Report

Title: “Cortical plasticity and interneuron recruitment in adolescents born to women with gestational diabetes mellitus”

In this work the authors aimed to determine whether altered I-wave recruitment was associated with neuroplastic responses in adolescents born to women with GDM. 20 GDM-exposed adolescents and 10 controls (aged 13.1 ± 1.0 years) participated. The authors used cTBS to induce neuroplasticity, while I-wave recruitment was assessed by comparing motor-evoked potential latencies using different TMS coil directions. In addition, recruitment of late I-waves was associated with stronger LTD-like neuroplastic responses to cTBS (p = < 0.001, R2 = 0.36). There were no differences between groups in mean neuroplasticity (p = 0.37), I-wave recruitment (p = 0.87), or the association between these variables (p = 0.41). However, in this work is shown that the relationship between I-wave recruitment and the response to cTBS previously observed in adults is also present in adolescents, while it does not appear to be altered significantly by in utero GDM exposure. In summary, the exposure to GDM does not appear to significantly impair LTD-like synaptic plasticity or interneuron recruitment.

General comment: This work is well written and well organized. However, the manuscript is not totally convincing and some improvements are still possible in order to enhance its quality and impact.

Some specific comments:

lines: “However, a principal issue with neuroplasticity induction by brain stimulation is that the responses are often highly variable both within and between individuals, with a number of known contributing factors, including age, cortisol, genetics, and prior muscle activity [10].”

*) Perhaps the authors could better explain this lines and the main consequences on the reliability of their results.

Lines: “20 GDM-exposed subjects (aged 12.7 ± 0.8 years [mean ± SD], 10 female),”

*) Perhaps this sample could be not too significant to drive relevant conclusion.. could the authors comment this point ? In other words, it is not too clear the reason of a this quite complex study performed with a so small sample of people. Please clarify.

Lines: “Given that we previously observed both weaker and more variable neuroplastic responses to cTBS (including more LTP-like responses) in children exposed to GDM when compared with a control group, we aimed in the present study to determine whether I-wave recruitment differed in this group and whether this could explain the increased variability in neuroplastic responses we observed. Given that the association between I-waves and neuroplasticity has only been studied in adults, our results also provide novel evidence for the presence of this effect in adolescents.”

together with:

Lines: “We have provided evidence indicating that the association between interneuron recruitment and the neuroplastic response to cTBS, whereby individuals who exhibit motor responses produced by late I-wave-generating circuitry have stronger and more predictable neuroplastic responses, is present in adolescents to a similar degree to that previously observed in adults [11-13]. “

and

Lines:” Thus, it does not appear that in utero GDM exposure fundamentally alters the relationship between I-wave recruitment and the neuroplastic response to cTBS.

and

However, in contrast to our previous study [5], we did not observe any systematic difference in response to cTBS in the GDM group compared with the control group.

and

However, the effect size in the group comparison is likely smaller than originally estimated and it may require a larger sample size to make reliable conclusions.

and

Given these results, we were unlikely to find support for our original hypothesis that differential I-wave recruitment would explain the difference in neuroplasticity between groups that we previously observed.

and

Responses in this range appear to have lower predictive value of the neuroplastic response to cTBS, and the association between I-wave recruitment and neuroplastic response therefore appeared weaker (and non-significant) in the GDM group when compared with the robust association seen in the control group. However, this comparison did not prove to be significant and we must therefore conclude that there is no difference in interneuron recruitment by TMS, or in the relationship between late I-wave recruitment and neuroplastic responses to cTBS, in children exposed to GDM, unless such an effect is smaller than we can detect with our sample size.

and

Thus, variation in the association between I-wave recruitment and response to cTBS may explain the apparent (albeit non-significant) difference in this relationship between groups seen in this study, rather than any intrinsic difference in physiology. Nevertheless, future research may extend this analysis with a larger sample to determine whether GDM-exposed individuals exhibit differences in the excitability or structure of I-wave-generating circuitry. In contrast to our previous study [5], we did not observe lower cortisol concentrations in the GDM group.

*) Starting from these lines the interested readers can not judge about the relevance of this work.

Indeed, correlations, when found, seem to be too weak together with a quite small sample size.

Could the authors better clarify the real value of this work also in comparison to the current literature ?

Lines:” In Chapter 5, we examine the cortisol awakening response and diurnal cortisol profile in children born to women with GDM “

*) Could the authors better explain the meaning of these lines ? What is the “Chapter 5” ?

Lines:”

Author Contributions: For research articles with several authors, a short paragraph specifying their individual contributions must be provided. The following statements should be used “Conceptual-

434 ization, X.X. and Y.Y.; methodology, X.X.; software, X.X.; validation, X.X., Y.Y. and Z.Z.; formal

435 analysis, X.X.; investigation, X.X.; resources, X.X.; data curation, X.X.; writing—original draft prep-

436 aration, X.X.; writing—review and editing, X.X.; visualization, X.X.; supervision, X.X.; project ad-

437 ministration, X.X.; funding acquisition, Y.Y. All authors have read and agreed to the published ver-

438 sion of the manuscript.” Please turn to the CRediT taxonomy for the term explanation. Authorship

439 must be limited to those who have contributed substantially to the work reported.

440 Funding: Please add: “This research received no external funding” or “This research was funded by

441 NAME OF FUNDER, grant number XXX” and “The APC was funded by XXX”. Check carefully that

442 the details given are accurate and use the standard spelling of funding agency names at

443 https://search.crossref.org/funding. Any errors may affect your future funding.

444 Institutional Review Board Statement: In this section, you should add the Institutional Review

445 Board Statement and approval number, if relevant to your study. You might choose to exclude this

446 statement if the study did not require ethical approval. Please note that the Editorial Office might

447 ask you for further information. Please add “The study was conducted according to the guidelines

448 of the Declaration of Helsinki, and approved by the Institutional Review Board (or Ethics Commit-

449 tee) of NAME OF INSTITUTE (protocol code XXX and date of approval).” OR “Ethical review and

450 approval were waived for this study, due to REASON (please provide a detailed justification).” OR

451 “Not applicable” for studies not involving humans or animals.

452 Informed Consent Statement: Informed written consent was obtained from parents/primary care-

453 givers. Assent was obtained from all the adolescent participants involved in the study.

454 Data Availability Statement: Please refer to suggested Data Availability Statements in section

455 “MDPI Research Data Policies” at https://www.mdpi.com/ethics.

456 Acknowledgments: In this section, you can acknowledge any support given which is not covered

457 by the author contribution or funding sections. This may include administrative and technical sup-

458 port, or donations in kind (e.g., materials used for experiments).

459 Conflicts of Interest: Declare conflicts of interest or state “The authors declare no conflict of inter-

460 est.” Authors must identify and declare any personal circumstances or interest that may be per-

461 ceived as inappropriately influencing the representation or interpretation of reported research re-

462 sults. Any role of the funders in the design of the study; in the collection, analyses or interpretation

463 of data; in the writing of the manuscript, or in the decision to publish the results must be declared

464 in this section. If there is no role, please state “The funders had no role in the design of the study; in

465 the collection, analyses, or interpretation of data; in the writing of the manuscript, or in the decision

466 to publish the results”.

*) Perhaps the authors should complete also these parts of the manuscript by inserting relevant data…

Author Response

Reviewer 2:

In this work the authors aimed to determine whether altered I-wave recruitment was associated with neuroplastic responses in adolescents born to women with GDM. 20 GDM-exposed adolescents and 10 controls (aged 13.1 ± 1.0 years) participated. The authors used cTBS to induce neuroplasticity, while I-wave recruitment was assessed by comparing motor-evoked potential latencies using different TMS coil directions. In addition, recruitment of late I-waves was associated with stronger LTD-like neuroplastic responses to cTBS (p = < 0.001, R2 = 0.36). There were no differences between groups in mean neuroplasticity (p = 0.37), I-wave recruitment (p = 0.87), or the association between these variables (p = 0.41). However, in this work is shown that the relationship between I-wave recruitment and the response to cTBS previously observed in adults is also present in adolescents, while it does not appear to be altered significantly by in utero GDM exposure. In summary, the exposure to GDM does not appear to significantly impair LTD-like synaptic plasticity or interneuron recruitment.

General comment: This work is well written and well organized. However, the manuscript is not totally convincing and some improvements are still possible in order to enhance its quality and impact.

Some specific comments:

  1. lines: “However, a principal issue with neuroplasticity induction by brain stimulation is that the responses are often highly variable both within and between individuals, with a number of known contributing factors, including age, cortisol, genetics, and prior muscle activity [10].”

*) Perhaps the authors could better explain this lines and the main consequences on the reliability of their results.

We are not entirely sure exactly what more the reviewer requires.  We have pointed the reader to a review of this topic (reference 10), and the methods describe how we’ve dealt with these factors (where possible), i.e., matching and recording age, measuring cortisol, limiting and standardising prior muscle activity and, of course, examining I-wave recruitment. These factors are then all discussed in relation to our results in the Discussion.

  1. Lines: “20 GDM-exposed subjects (aged 12.7 ± 0.8 years [mean ± SD], 10 female),”

*) Perhaps this sample could be not too significant to drive relevant conclusion.. could the authors comment this point ? In other words, it is not too clear the reason of a this quite complex study performed with a so small sample of people. Please clarify.

We have explained at a number of points in the manuscript that we acknowledge the relatively small sample size. It is one of the difficulties of trying to recruit birth cohort participants 12 years after their initial recruitment. This group have also been quite heavily studied as part of the original randomised controlled trial. The conclusion is that the relationship between interneuron recruitment and neuroplastic response to cTBS does not differ between the GDM and control groups. It is explained clearly in the discussion (line 402) that because the effect was very strong in the control group – and would decrease with increasing sample size – the data are biased towards a false positive, and therefore a negative result (as we found) is unlikely to be a product of the small sample. Increasing the size of the control sample would likely not alter the conclusions or indeed add significant value to the study. Increasing the size of the GDM group might, be we were strictly limited by the number of GDM-exposed children examined in the first study (and ultimately in the MiG cohort) – and we recruited every one that we could.

  1. Lines: “Given that we previously observed both weaker and more variable neuroplastic responses to cTBS (including more LTP-like responses) in children exposed to GDM when compared with a control group, we aimed in the present study to determine whether I-wave recruitment differed in this group and whether this could explain the increased variability in neuroplastic responses we observed. Given that the association between I-waves and neuroplasticity has only been studied in adults, our results also provide novel evidence for the presence of this effect in adolescents.” 

together with:

Lines: “We have provided evidence indicating that the association between interneuron recruitment and the neuroplastic response to cTBS, whereby individuals who exhibit motor responses produced by late I-wave-generating circuitry have stronger and more predictable neuroplastic responses, is present in adolescents to a similar degree to that previously observed in adults [11-13]. “

and

Lines:” Thus, it does not appear that in utero GDM exposure fundamentally alters the relationship between I-wave recruitment and the neuroplastic response to cTBS.

and

However, in contrast to our previous study [5], we did not observe any systematic difference in response to cTBS in the GDM group compared with the control group.

and

However, the effect size in the group comparison is likely smaller than originally estimated and it may require a larger sample size to make reliable conclusions.

and

Given these results, we were unlikely to find support for our original hypothesis that differential I-wave recruitment would explain the difference in neuroplasticity between groups that we previously observed.

and

Responses in this range appear to have lower predictive value of the neuroplastic response to cTBS, and the association between I-wave recruitment and neuroplastic response therefore appeared weaker (and non-significant) in the GDM group when compared with the robust association seen in the control group. However, this comparison did not prove to be significant and we must therefore conclude that there is no difference in interneuron recruitment by TMS, or in the relationship between late I-wave recruitment and neuroplastic responses to cTBS, in children exposed to GDM, unless such an effect is smaller than we can detect with our sample size.

and

Thus, variation in the association between I-wave recruitment and response to cTBS may explain the apparent (albeit non-significant) difference in this relationship between groups seen in this study, rather than any intrinsic difference in physiology. Nevertheless, future research may extend this analysis with a larger sample to determine whether GDM-exposed individuals exhibit differences in the excitability or structure of I-wave-generating circuitry. In contrast to our previous study [5], we did not observe lower cortisol concentrations in the GDM group.

*) Starting from these lines the interested readers can not judge about the relevance of this work. Indeed, correlations, when found, seem to be too weak together with a quite small sample size. Could the authors better clarify the real value of this work also in comparison to the current literature ?

We are at a bit of a loss to understand exactly what the reviewer is asking of us. We have explained at a number of points in the manuscript that we acknowledge the relatively small sample size. It is one of the difficulties of trying to recruit birth cohort participants 12 years after their initial recruitment. We think that the paragraph (starting at line 386) is very clear about how our findings can be interpreted, with relevance to the sample size, but also, perhaps more importantly, to the effect size.

  1. Lines:” In Chapter 5, we examine the cortisol awakening response and diurnal cortisol profile in children born to women with GDM “

*) Could the authors better explain the meaning of these lines ? What is the “Chapter 5” ?

This error has been corrected